# Templated bilayer self-assembly of fully conjugated π-expanded macrocyclic oligothiophenes complexed with fullerenes

José D. Cojal González[1], Masahiko Iyoda[2] & Jürgen P. Rabe[1]

Fully conjugated macrocyclic oligothiophenes exhibit a combination of highly attractive structural, optical and electronic properties, and multifunctional molecular thin film architectures thereof are envisioned. However, control over the self-assembly of such systems becomes increasingly challenging, the more complex the target structures are. Here we show a robust self-assembly based on hierarchical non-covalent interactions. A self-assembled monolayer of hydrogen-bonded trimesic acid at the interface between an organic solution and graphite provides host-sites for the epitaxial ordering of Saturn-like complexes of fullerenes with oligothiophene macrocycles in mono- and bilayers. STM tomography verifies the formation of the templated layers. Molecular dynamics simulations corroborate the conformational stability and assign the adsorption sites of the adlayers. Scanning tunnelling spectroscopy determines their rectification characteristics. Current–voltage characteristics reveal the modification of the rectifying properties of the macrocycles by the formation of donor–acceptor complexes in a densely packed all-self-assembled supramolecular nanostructure.

[1] Department of Physics & IRIS Adlershof, Humboldt-Universität zu Berlin, D-12489 Berlin, Germany. [2] Department of Chemistry, Graduate School of Science and Engineering, Tokyo Metropolitan University, Hachioji, Tokyo 192-0397, Japan. Correspondence and requests for materials should be addressed to J.P.R. (email: rabe@physik.hu-berlin.de).

Control over the self-assembly of molecular structures on a solid surface is of particular interest towards three-dimensional (3D) engineering of supramolecular materials[1] and designing of molecular electronic devices[2–4]. Hydrogen-bonding is the most versatile, yet specific type of weak interaction to engineer two-dimensional (2D) self-assembled networks[5,6] due to the high level of control at the molecular level[7,8] and the potential applications[9]. Among other supramolecular interactions, host–guest complexation on 2D networks provides an interesting platform to create functional multicomponent layers[10,11], nanostructures[12,13] and prototypical devices[14], and establish the first step to tailor 3D architectures[15,16]. Furthermore, donor (D) - acceptor (A) complexation of common acceptor molecules into the core of giant macrocycles unveils promising building blocks for the fabrication of complex nanostructures[15,17–19] and self-assembled devices. The design of multifunctional organic materials containing fullerene $C_{60}$ as an active component has been extensively investigated in organic electronics[20,21]. The spherical geometry of fullerene $C_{60}$ and its acceptor properties render a cylindrical or torus-like donor molecule a promising choice to act as a complementary host-site[22,23]. Similar to its linear counterparts[24,25], π-expanded macrocyclic oligothiophenes have attracted attention for their potential applications in molecular electronics and as components in molecular devices[15,17,26,27].

Here we show how to create a bilayer of moderately rectifying donor–acceptor complexes via double hosting of $C_{60}$ molecules. This architecture is achieved by combining the spontaneous formation of a hydrogen-bonded trimesic acid (TMA, **1**) as a 2D-template at the interface between its heptanoic acid solution and a highly oriented pyrolytic graphite (HOPG) substrate[12,28], the complexation of oligothiophene macrocycles[29,30] and $C_{60}$ in solution, and the site recognition of the cavities in the TMA monolayer by $C_{60}$ (ref. 31). Moreover, we demonstrate the potential of this multi-step articulated self-assembled construction to engineer multiple epitaxially grown macrocycle layers.

## Results

**2D-template of trimesic acid**. Previous experiments using π-expanded oligothiophenes, that is, macrocycles composed of six[32] and eight thiophene units (E,E-8-mer, **2**) (ref. 30), have shown that the self-assembly of an ordered monolayer is favourable at the interface between HOPG and a fatty acid solution, where under similar experimental conditions $C_{60}$ molecules alone do not adsorb[33]. Here, we found that the attractive interaction between the HOPG substrate and **2** is not sufficient to drive the formation of an ordered monolayer of the Saturn-like complex (E,E-8-mer · $C_{60}$, **3**) at the interface between its heptanoic acid solution and the graphite surface, which may be due to the most favourable energy interaction of the complex. Therefore, we looked for a modification of the substrate to adsorb both components of the complex, macrocycle and $C_{60}$.

A hydrogen-bonding driven 2D network of TMA has been considered as a template for the complexes due to its capability to host small molecules, such as $C_{60}$ (ref. 31). Fullerenes $C_{60}$ can be hosted within the honeycomb-like network of TMA, mainly due to the favourable energetic stabilization of the host–guest interaction, as calculated previously using all force-field methods[31]. Here we report the recognition of the host-sites in a previously adsorbed self-assembled monolayer of TMA by the in-solution complexed $C_{60}$ molecules.

As depicted in Fig. 1a,b, TMA forms an extended honeycomb-like structure when adsorbed at the solid–liquid interface between its solution in heptanoic acid and HOPG. The hexagonal cavities of the periodic arrangement are created by the di-apto

hydrogen-bonding of adjacent carboxylic acid groups of six neighbouring molecules, exhibiting a pore diameter of about 12 Å (refs 12,28). Figure 1c shows a scanning tunnelling microscopy (STM)[34] height image of the chicken-wire structure of the 2D TMA-crystal on HOPG upon the addition of a small drop of saturated heptanoic acid solution; the unit cell parameters agree with previous reports[12,31].

**Self-assembly of E,E-8-mer · $C_{60}$ on TMA template**. We schemed complex **3** (Fig. 1d) to form an epitaxial monolayer with the TMA network as sketched in Fig. 1e, displaying **3** without butyl groups for clarity. The addition of a drop of heptanoic acid solution of **3** to the previously formed honeycomb TMA network causes the spontaneous formation of a hexagonal network with one molecular complex per unit cell (Fig. 1f). Steric hindrance prevents two neighbouring pores to host one complex each; every complex **3** is rather hosted in an alternate pore of the TMA honeycomb, making the unit cell of **3** a ($\sqrt{3} \times \sqrt{3}$)R30° superstructure of the unit cell of TMA. In the STM image (Fig. 1f) one finds donut-like shaped objects, where the tunnelling current is larger compared with surrounding regions[35]. This effect is observed at positive sample bias ($+0.70\,V < U_s < +1.20\,V$), whereas at negative bias the image of the TMA honeycomb-like network is recovered (details in section STM tomography). Contrastingly, when a drop of a heptanoic acid solution of the macrocycle **2** is added to the previously formed honeycomb TMA network, an irregular aggregation of the molecules is observed (Supplementary Fig. 1), suggesting that the formation of an ordered structure depends critically on the presence of the TMA hosting sites along with the $C_{60}$ molecule of the complex.

Two possible pathways could afford the formation of the templated monolayer of **3**: a two-step adsorption, in which non-complexed $C_{60}$ is adsorbed on TMA vacant sites followed by the formation of the complex **3** again on these TMA sites, and a one-step adsorption, where in-solution complex **3** directly gets adsorbed on the TMA sites. The low solubility of $C_{60}$ in heptanoic acid and its low density host–guest complexation on a TMA honeycomb network[31], favours the one-step adsorption process.

Over large scales the templated monolayer is characterized by single crystalline domains with diameters up to a few 10 nm, separated by grain boundaries, which are mostly slip lines (Supplementary Fig. 2). This observation indicates that the template is not perfectly rigid but somewhat soft. Using the data from the fast fourier transform of Supplementary Fig. 2, we found that the unit cell parameters of the macrocycle network follow the relations: $a_2 = \sqrt{3} \cdot a_1$, $b_2 = \sqrt{3} \cdot b_1$ and $\theta_1 = \theta_2 = 60°$, where $a_1$, $b_1$ and $\theta_1$ are the unit cell parameters of the TMA network. Taking $a_1 = b_1 = 1.7$ nm, one gets $a_2 = b_2 = 2.9$ nm, which is in perfect accordance with the measured results (including the data presented in Fig. 1f). The maximum length of a complex **3** with fully stretched butyl groups is about 3.2 nm, according to X-ray crystallographic data[30]. This length is larger than its unit cell parameters on the TMA network and since STM images could not resolve the butyl substituent groups, we assume the side chains of **3** are either back-folded into the supernatant solution or into the empty pores of the template (see molecular dynamics section below), which might be attributed to the incommensurability between the hexagonal TMA-networks and the quasi-eightfold symmetry of **3**.

**STM tomography**. Figures 1f and 2b display STM height images after the formation of the templated monolayer of **3**. At currents on the order of 50 pA, stable images could be only recorded at positive sample bias, while at negative sample bias the TMA template was

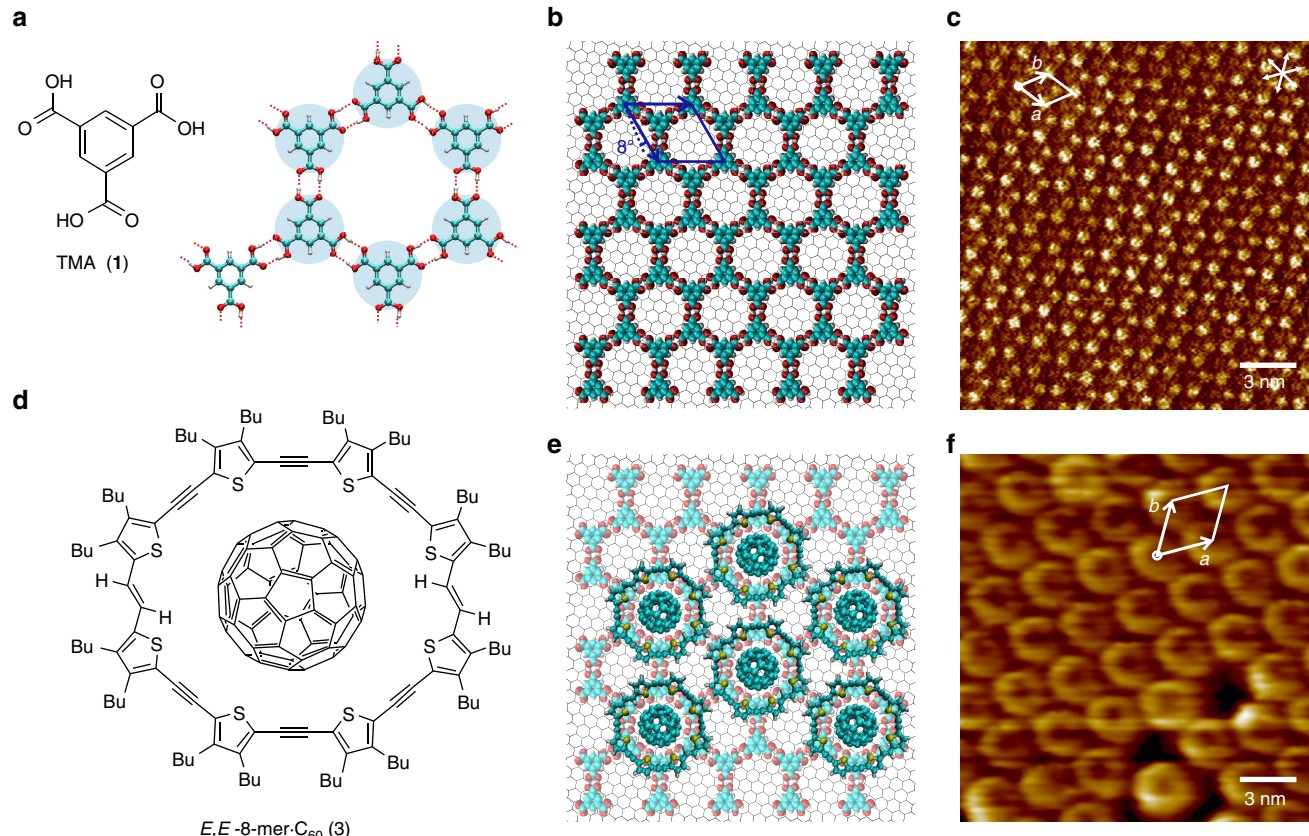

**Figure 1 | 2D-template of TMA for a π-expanded macrocycle complex with $C_{60}$.** (**a**) Each TMA molecule forms hydrogen-bonding with any three neighbour molecules rotated by 60°, 120° and 180°. (**b**) Section of the honeycomb-like network formed on the basal plane of HOPG as shown schematically, the size of the host cavity is around 12 Å. The hexagonal unit cell ($a = b = 1.70$ nm, $\theta = 60°$) is rotated 8° respect to the HOPG axis. (**c**) STM height image of a honeycomb-like network of TMA, unit cell $a = (1.70 \pm 0.09)$ nm, $b = (1.69 \pm 0.03)$ nm, $\theta = (60 \pm 3)°$, $U_s = -1.01$ V, $I_t = 46$ pA. (**d**) Molecular structure of the π-expanded oligothiophene macrocycles in its $E,E$ form complexed with $C_{60}$. Fullerenes $C_{60}$ are coordinated by the electron donor structure of the macrocycle. (**e**) Molecular schematic of the epitaxially arranged **3** monolayer, where $C_{60}$ is hosted in the pores of the TMA-template network. (**f**) STM height image of the 2D network formed upon the addition of a drop of solution of **3** after forming the template in **c**. Unit cell: $a = (2.84 \pm 0.14)$ nm, $b = (2.83 \pm 0.15)$ nm, $\theta = (62 \pm 5)°$, $U_s = +0.90$ V, $I_t = 50$ pA.

observed (Fig. 2c); at lower negative as well as positive bias (smaller than 0.6 V) no structures were recognized; and at high negative bias (−1.25 V) unstable donut-like molecules were observed (Fig. 2d and Supplementary Fig. 3). Moreover, $C_{60}$ molecules could not be identified at any substrate bias up to 1.2 V in both polarities, while in similar templated systems ($C_{60}$ on cyclothiophene[17], TMA[31] or $p$-terphenyl-3,5,3″,5″-tetracarboxylic acid, TPTC[16]) they appeared as high-contrast spots. This indicates that in the present case $C_{60}$ molecules are either absent or not visible. The experimental results reported above can be explained on the basis of a STM tomography model[36]. The model assumes that the energy frontier orbitals involved in the resonant tunnelling through the molecules within the tip-HOPG gap are shifted proportional to their distance to the surface, mainly due to its asymmetric alignment[37]. Within the model, the applied bias affects the visibility of the TMA template and the complex **3**.

Figure 2a shows the position of the highest occupied molecular orbital (HOMO) of **3**, located at the macrocycle, and the lowest unoccupied molecular orbital (LUMO), which lies on the $C_{60}$, as deduced from gas-phase density functional theory calculations on unsubstituted **3** (Supplementary Fig. 4 and Supplementary Methods).

Figure 2b–d depict energy diagrams of the position of the HOMO (see Supplementary Fig. 5) of the TMA template (H1) and complex **3** (H2) within the tunnelling region for positively biased substrate (Fig. 2b) and negatively biased substrate

(Fig. 2c,d), respectively. Empty rectangles show the position of the HOMO when no bias is applied, while filled rectangles are the shifted orbital levels.

As depicted in Fig. 2b, when the substrate is positively biased the resonance effect occurs as the electrons from the tip tunnel through the HOMO of the complex **3** (H2), which is shifted upwards (positive energy) more than the HOMO of the TMA (H1). The STM height image shows the donut-like shape of **3** which is associated with H2. At negative sample bias (Fig. 2c), the lower position (more negative energy) of H1 makes it more accessible at negative imaging bias (between −0.7 and −1.1 V); here the STM height image depicts the honeycomb-like network of TMA. In addition, lowering the voltage even more (Fig. 2d) causes H2 to shift more downwards (negative energy), which makes it accessible again at bias around −1.25 V, but unstable for steady imaging.

Moreover, in the case where the substrate is positively biased, the LUMO of the complex (not shown in Fig. 2) shifts upwards (positive energy), and when it is negatively biased the shift downwards (negative energy) is not sufficient to be in resonance due to the distance of the LUMO to the Fermi-level of HOPG, thus making the $C_{60}$ invisible in both cases.

**Scanning tunnelling spectroscopy.** Scanning tunnelling spectroscopy (STS) has been used to determine the local electronic

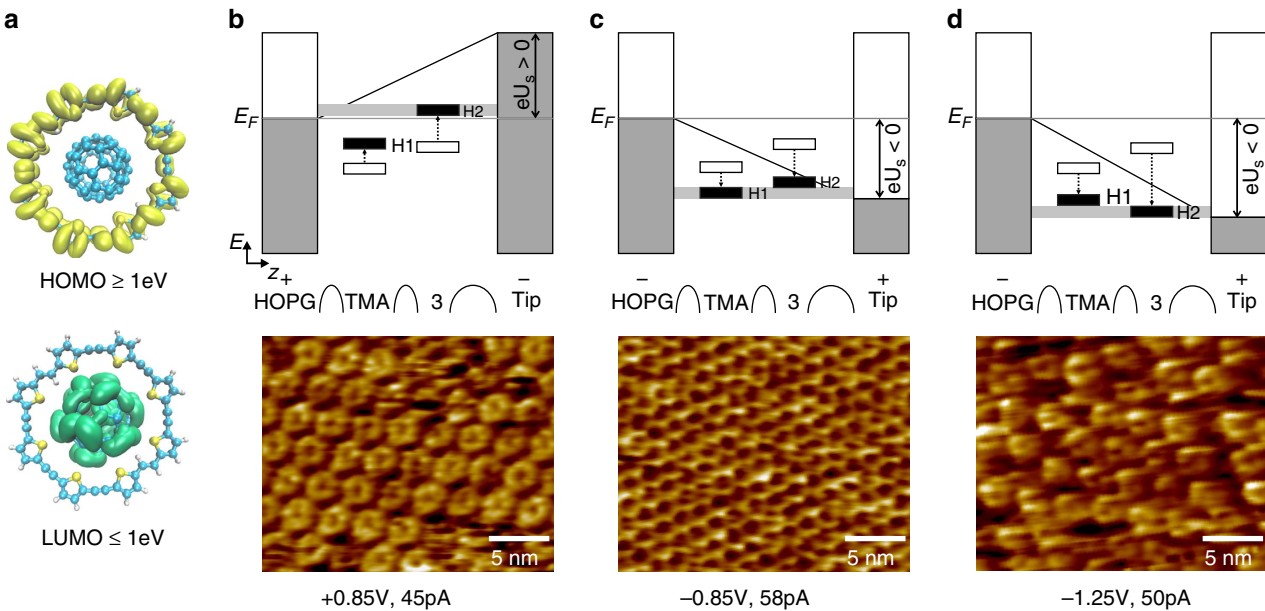

**Figure 2 | Bias set-point imaging.** (**a**) Isosurfaces showing the electron probability distribution 1 eV below and above the HOMO and LUMO of the complex **3**, respectively. (**b**) At positive substrate bias, the resonant tunnelling is achieved through the HOMO of **3** (H2), which is symmetrically distributed in the macrocycle. The height image shows below at $U_s = +0.85$ V, $I_t = 45$ pA. (**c**) By switching to negative substrate bias, the resonant tunnelling occurs through the HOMO of the underneath TMA (H1), as the image below at $U_s = -0.85$ V, $I_t = 58$ pA. (**d**) Lowering the negative substrate bias gives access again to H2, $U_s = -1.25$ V, $I_t = 50$ pA. The tunnelling does not occur through the LUMO of **3** (localized in the $C_{60}$) because of the closer proximity of both HOMOs to the Fermi level of the HOPG.

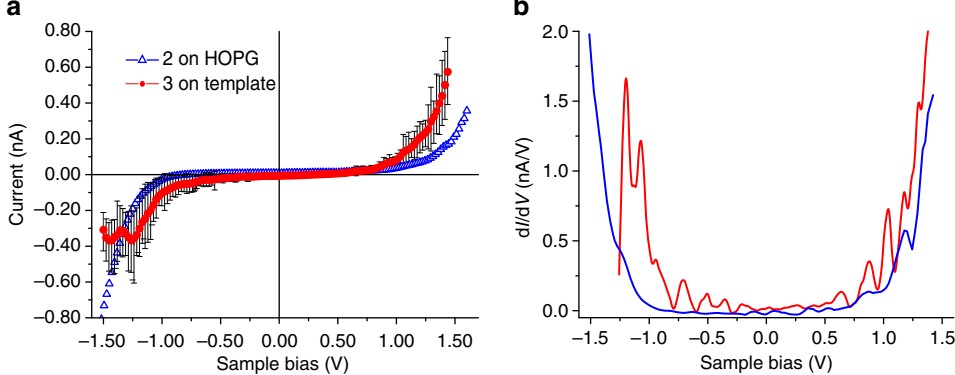

**Figure 3 | STS of 2 and templated monolayer of 3.** (**a**) Average $I$–$V$ characteristics over twenty measurements of a monolayer of templated **3** (red circles, STS conditions $U_s = +0.94$ V and $I_t = 50$ pA), including error bars (see Supplementary Fig. 6a for scattered data). For comparison the average $I$–$V$ trace of fifteen measurements of **2** on HOPG was measured too (blue triangles, STS conditions $U_s = -1.0$ V and $I_t = 80$ pA). For the templated monolayer of **3** the current saturates below $-1.2$ V, not the case for **2**. (**b**) Differential conductance ($dI/dV$) versus sample bias, where the saturation effect is evident below $-1.2$ V.

structure of the templated monolayer(s) of **3**. Figure 3a depicts the current–voltage ($I$–$V$) characteristics across the macrocycle **2** on HOPG and the templated complex **3**. The $I$–$V$ characteristics of **2** exhibits an asymmetric behaviour with a larger tunnelling current at negative sample bias, showing the expected rectification behaviour of a donor molecule[38]. On the contrary, the $I$–$V$ characteristics through the donut-like moiety of **3** shows an asymmetric behaviour, with a larger tunnelling current at positive sample bias. The charge transport across the center of the templated complex, where the $C_{60}$ is located, is symmetric (Supplementary Fig. 6b).

Moreover, for the case of the transport across the rim of **3**, there is a saturation in the current at $-(0.35 \pm 0.12)$ nA in a bias range between $-1.2$ and $-1.5$ V; similar findings have been previously reported for thiophene macrocycle networks covered

with $C_{60}$ (ref. 17). This tendency is dramatically shown in the $dI/dV$ spectra (Fig. 3b), which reflects more directly the local density of states of the system[39]. In the range of $-1.5$ to $+1.5$ V the charge transport across **2** increases monotonously at both positive and negative bias, while for complex **3** a blockage at $-1.2$ V is observed, which is attributed to the HOMO. A correspondingly clear feature of the LUMO is, however, not observed in the here accessible bias range up to $+1.4$ V. The saturation of the current at negative bias values is associated with delocalization of excited states in the infinite $\pi$-conjugated system of the macrocycle[40] and represents a key spectroscopic characteristic of this type of systems[17].

By creating the templated monolayer of **3**, in which the presence of the $C_{60}$ molecules is essential for a regular array of the macrocycles epitaxial to the TMA network, the formation of

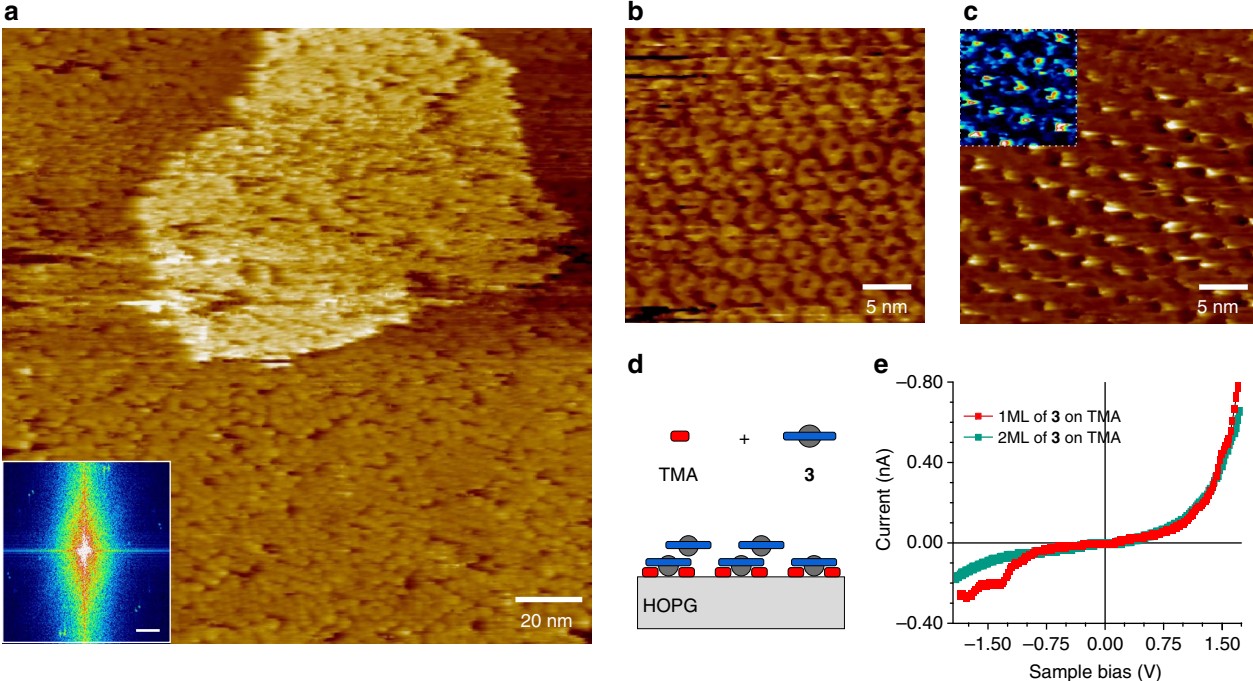

**Figure 4 | Growth of a second monolayer of 3. (a)** STM height image of a templated bilayer of **3** in heptanoic acid solution. Measuring conditions $U_s = +1.05$ V, $I_t = 42$ pA. Two different unit cells can be identified from the fast fourier transform in the inset, the scale bar is 1/10 nm. STM height image at $+1.05$ V (**b**) and $-1.05$ V (**c**) of a templated bilayer (2ML) of **3** in heptanoic acid solution. The unit cell measures $a = (2.85 \pm 0.04)$ nm, $b = (2.84 \pm 0.04)$ nm, $\theta = (61 \pm 1)°$. An inset in **c** is showed in a different contrast to highlight the image of the macrocycles of the first templated monolayer (1ML). We attribute the bright spot to the $C_{60}$ of the layer on top. (**d**) Scheme of the suggested arrangement of the first and second layer of **3**. $C_{60}$ of the second layer lies on the rim of the first one. (**e**) $I$–$V$ characteristic over fifteen measurements through 1ML and 2ML of **3** (STS conditions $U_s = +0.95$ V and $I_t = 51$ pA) (see Supplementary Fig. 9 for scattering of data).

rectifying monolayer of complexes is accomplished and the rectification behaviour is likewise inversed.

**Templated bilayer of Saturn-like complexes**. The density of the templated monolayer of **3** increments with time (Supplementary Fig. 7). Saturation of the first monolayer is obtained by increasing the solution concentration. After some minutes, while keeping the imaging conditions, islands similar to the one depicted in Fig. 4a are formed. A histogram of Fig. 4a gives a relative height of 2 Å for the second layer of **3** (Supplementary Fig. 8). This step height is smaller than the typically observed for non-interacting $C_{60}$ measured by STM[41], nevertheless, the apparent height in STM experiments is related to the local density of states and, therefore, is not to taken as an absolute dimension. We attribute these islands to a second monolayer of the complex **3** grown epitaxial to the first monolayer, creating a templated bilayer **3|3**. Similar bilayers are known to occur by π–π stacking of polycyclic aromatic hydrocarbons[42], metallo-ligand interactions of phthalocyanines[43] and guest-induced of two layers of TPTC[16].

The arrangement of the second layer relative to the first one is resolved by applying set-point dependent imaging as described earlier for the template monolayer of **3**. The image recorded in Fig. 4b shows the top layer of **3** at positive bias. Upon switching the bias to negative values, an arrangement of bright spots and low contrast macrocycles is imaged (Fig. 4c), which we associate with $C_{60}$ molecules of the top layer of **3** sitting on the rims of the first layer complex. This adsorption site for $C_{60}$ has been reported to be the most favourable on other types of shape-persistent macrocycles[15,17].

STS measurements through one and two monolayers of **3** showed that the forward rectification is kept through the templated bilayer **3|3** (Fig. 4e). However, the saturation point seen across a monolayer, was not reached for values down to $-1.8$ V in the case of the bilayer. The rectification ratio, defined as $R(V_0) = |I(V_0)/I(-V_0)|$, where $I(V_0)$ is the current at a reference bias $V_0$ (typically the highest bias used), reaches an average value of $4.9 \pm 1.2$ for the templated bilayer **3|3** at 1.7 V, while for the monolayer of **3** it is $2.7 \pm 1.1$ at 1.7 V, thus improving the rectification upon addition of a second monolayer. For comparison, the average value of $R$ for a monolayer of **2** on HOPG is $3.5 \pm 1.1$. Average values of $R$ are the arithmetic mean and standard deviation for each curve in Supplementary Figs 6c and 9.

**Molecular dynamics simulation**. In order to rationalize the formation of bilayer architectures of **3** on the TMA network in the light of the results in Fig. 4b,c, all force-field calculations were performed using the program NAMD[44] with the CHARMM general force field[45] (CGenff). The multilayer was simulated in an orthorhombic box with periodic boundary conditions to reproduce an infinite system. First, we placed a twelve pore TMA honeycomb lattice (24 TMA molecules) 6 Å above the 52 Å × 60 Å graphene together with four **3** complexes at 6 Å form the center of every second pore of TMA (12 Å above graphene) (Supplementary Fig. 11), and then energy minimized while keeping the substrate fixed. A two nanosecond molecular dynamic simulation at 300 K shows the stability of the $(\sqrt{3} \times \sqrt{3})$R30° superstructure with a TMA-graphene separation of $\sim 3.45$ Å, the fullerenes $C_{60}$ $\sim 6.6$ Å from graphene and the macrocyclic part of **3** $\sim 0.6$ Å above the complexed $C_{60}$ molecules. The in-plane position of the complex is stabilized with its center of mass between 0.3 A and 0.9 A to the

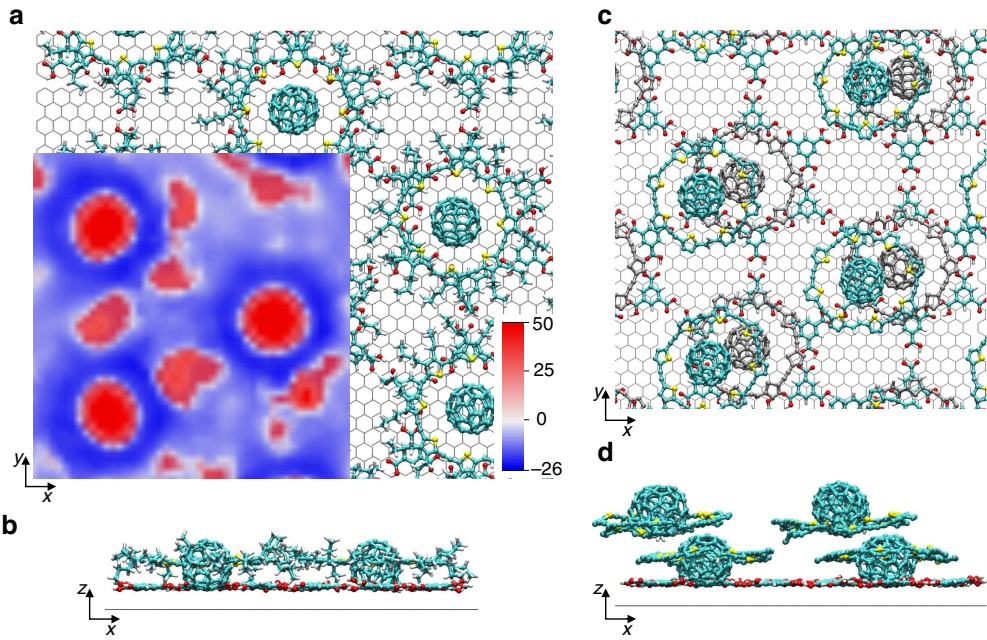

**Figure 5 | Structure of the templated bilayer 3|3|TMA|HOPG.** Snapshot of the self-assembled monolayer of **3** on the TMA template after 2 ns simulation, top (**a**) and side view (**b**). A potential mean force map of a templated monolayer of **3** probed with a $C_{60}$ averaged on 1 ns of molecular dynamic simulation (energy given in $kcal\,mol^{-1}$) is also shown. The positions coloured in blue are the most favourable for the molecule in agreement with the STM image in Fig. 4c. (**c,d**) Top and side view of the templated bilayer of **3** after 2 ns of molecular dynamic simulation at 300 K, the alkyl chains are omitted for clarity.

center of the pore. Moreover, Fig. 5b shows a side view of the templated monolayer, where the alkyl chains are folded up, down into the empty pores of the template and in between complexes, confirming our previous assumption.

Furthermore, we exploited implicit ligand sampling (ILS)[46] to find the most accessible and energetically favourable position of the second monolayer of molecules. This computational method evaluates the potential mean force (PMF) of placing a probe molecule everywhere close to the system by means of the Gibbs free-energy cost. Figure 5a shows the PMF map of probing with one $C_{60}$ at 13.5 Å averaged on 1 ns of molecular dynamic simulation at 300 K (25 frames), which locates the minimum energy positions where the probe molecule stands above the rim of the underlying molecule as it was suggested by the STM measurements in Fig. 4. Finally, four **3** complexes were placed on the positions of the ILS map minima forming a second monolayer (Supplementary Fig. 12) and a further two nanoseconds molecular dynamic simulation analysed the behaviour of the templated bilayer layer while keeping the substrate fixed. Figure 5c,d show snapshots after 2 ns of simulation, interestingly the macrocyclic part of **3** of the second layer lowers 1.7 Å with respect to the corresponding $C_{60}$ (12 Å from HOPG), while the position of the first templated layer is not affected. This positioning decreases the energy of the system in 35 $kcal\,mol^{-1}$ per complex **3** in the second layer (Supplementary Fig. 13), which grows linearly with the number of molecules involved in the assembly. The average of the trajectory snapshots of the molecular dynamic simulation under ideal feedback conditions shows that the unit cell for the first and second monolayer is kept within the experimental values (see Supplementary Fig. 14). Full detailed information about the molecular dynamics calculation is provided in Supplementary Note 2.

## Discussion
We demonstrated that a conjugated π-expanded macrocyclic oligothiophene hosting fullerenes $C_{60}$ can be self-assembled into the pores of hexagonally packed TMA to form three-component rectifying bilayers and triple layers with high densities of molecular diodes. The formation of the supramolecular nanostructure depends critically on the presence of both the TMA and the hosted $C_{60}$. We used bias dependent imaging and all-atom molecular dynamics calculations to establish the structure of the templated mono- and bilayers of complexes and their epitaxial arrangement with respect to the TMA network beneath. STS measurements indicate the inversion of the rectifying characteristics of the macrocycles in the templated monolayer. While templated monolayer of **3** is a very moderate rectifier, the addition of a second monolayer of **3** improves the rectification ratio. Our findings show that a combination of non-covalent interactions, including hydrogen-bonding and charge transfer in host-guest systems, allows supramolecular engineering of self-assembled functional nanosystems featuring ways to control their electronic properties.

## Methods

**STM measurements.** A home-built beetle-type scanning tunnelling microscope with an Omicron controller was used for all the experimental measurements. Small drops ($\sim 5\,\mu l$) of heptanoic acid solutions of the compounds ($10^{-4}$–$10^{-5}$ M) were applied to freshly cleavage HOPG substrates. The substrates were glued to a glass holder and electrically contacted using silver paste. STM tips were mechanically cut from a Pt/Ir wire (90/10, 0.25 mm diameter). The STM data were processed using SPIP software (Image Metrology A/S). Thermal drift was corrected with respect to the known hexagonal HOPG lattice underneath; background flattening and despeckling filter were also applied. The HOPG lattice was visualized by lowering the tunnelling voltage to 20 mV and raising the tunnelling current to 1–1.4 nA.

**STS measurements.** During STM imaging at constant current (feedback on), the STM tip was moved to the point of interest. Hence, the tip position was kept fixed according to the gap voltage and setpoint for the tunnelling current. The feedback loop was then switched off and a bias ramp was swept through and the tunnelling current recorded. In order to decrease the effect of lateral drift while recording STS, the tip was positioned over the region of interest as close as possible to the most recently scanned line. Typically, it takes about 25 ms to perform a single STS curve and the lateral drift velocity is $<0.1\,nm\,s^{-1}$. The actual tip-sample separation for a single measurement follows a normal distribution around the mean value of the

setpoint, thus the average of the data is obtained by a lognormal function to discard inappropriate data which could arise by fluctuations of the tip structure[47].

**Molecular dynamics calculations.** All simulations were performed using the NAMD software (2.10 build for linux), with the CHARMM general force field (CGenff) and parametrization of the molecule **2** (see Supplementary Note 1, Supplementary Fig. 10 and Supplementary Table 1) using the force field toolkit[48] implemented in VMD visualization software[49]. The graphene slab substrate was kept fixed on all simulations, its atoms were assigned atom type CG2R61 and no partial charge. A Langevin thermostat with a damping coefficient of $1 \, ps^{-1}$ and a time step of 2 fs was used in all simulations. A distance cutoff of the size of the periodic box was used to compute non-bonded interactions. Electrostatics interactions were calculated using the Particle mesh Ewald method with a space grid of 0.5 Å. For the ILS calculation, periodic boundary condition were applied, with a resolution (space grid) of 1 Å and a subsampling of 3 points. The trajectory snapshots for PMF map calculation were obtained after extra 1 ns simulation, were both TMA and graphene were kept fix.

**Data availability.** The authors declare that the data supporting the findings of this study are available within the article and its Supplementary Information file or from the corresponding authors upon reasonable request.

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

## Acknowledgements

We thank C.-A. Palma (TU München) and N. Severin (HU Berlin) for fruitful discussion. This work was supported by the German Science Foundation (DFG) SFB 765 and Ra482/6-1 and JST, Strategic International Research Cooperative Program (SICP) [Strategic Japanese-German Cooperation Program from JST].

## Author contributions

J.D.C.G. acquired the STM data and performed the molecular dynamic simulations, M.I. developed the synthetic route for the macrocycle **2** and complex **3**, J.D.C.G. and J.P.R. analysed the data, J.D.C.G. and J.P.R. conceived and coordinated the experimental work, J.P.R and M.I. coordinated the project and J.D.C.G. and J.P.R. co-wrote the paper with inputs from M.I.

## Additional information

**Competing financial interests:** The authors declare no competing financial interests.

**Publisher's note**: 

