## [Peer Review File · Nature Communications]

Reviewers' comments:

Reviewer #1 (Remarks to the Author):

This study is one of the very few reports on the supramolecular organization of multicomponent functional molecules. A self-assembled monolayer of trimesic acid at the liquid (1-heptanoic acid) – solid (graphite) interface acts as a template to host a complex of an oligothiophene macrocycle and C60. The TMA layer is necessary to direct the self-assembly of the oligothiophene macrocycle – C60 complex. The subsequent organisation of this complex is ruled by the crystallinity of the TMA template layer. Molecular dynamics simulations confirm the tentative models based on the STM imaging. Furthermore, evidence for multilayer formation is provided. Interestingly, using scanning tunneling spectroscopy, space-resolve info is provided on the local electronic characteristics of this multicomponent system, showing rectifying behavior. A second layer of the macrocycle – C60 complex improves the rectifying behaviour.

While several aspects of this study have in one way or the other been reported before (not necessarily for the same system), e.g.

- Complexation of C60 and an oligothiophene macrocycle (Bauerle et al.)
- STS on such complex of C60 and an oligothiophene macrocycle (Bauerle et al.)
- STS on multicomponent systems (by Rabe et al.)
- Multicomponent self-assembly involving trimesic acid (Lackinger et al., De Feyter et al.)
- Templating effect of trimesic acid
- Multilayer formation involving C60 (Beton et al.)

the combination provided in this study is of interest, of high quality, and complex. The data (STS) and data interpretation are supported by semi-quantitative modelling, based on concepts demonstrated by the Rabe group before.

While I favour publication, a number of aspects should be addressed in detail. In particular, more details should be given (in the supporting information). At this stage, not enough information is given to evaluate the statistical relevance of the STS data. In addition, certain statements seem not very well supported by the experimental evidence (e.g. difference in rectification ratio between 1 ML and 2 ML)

- Concentration / composition of the solutions (for STS and STM topographs).

- Details on the STS

o Fig. 3: what are the error bars for positive biases?

o In general: What is the initial setpoint (current, voltage) to record the STS curves? (provide in Caption Figure 4, not in Figure 3).

o Fig. 3: Why a "saturation" at -1,2 V? What is meant by "saturation"? Do you mean "levels off"?

o Fig. S7: are these averaged curves?

o Page 6, supporting info: what do you mean by "adequate average"?

o What about lateral drift effects during STS recording? In other words, how reliable are the claims concerning the exact position of the STS recording (e.g. center of the macrocycles, etc.)

o In general, STS data presented: are these recorded in the same session? What about differences between sessions?

o Fig S9b: rectification ratio: based upon these data, it is dangerous to claim there is a significant difference in the rectification ratio between 1 ML and 2 ML, ad the data are very scattered.

In addition, there are several minor typos that should be corrected.

Reviewer #2 (Remarks to the Author):

The article by Rabe et al. builds upon a previous study (reference 30) which describes the synthesis and characterisation of the 'Saturn-like' complexes formed between oligothiophenes and C60. In the current study a series of additional surface based characterisation is reported including STS etc.. The work is well carried out and it is difficult to criticise the quality of the science.

The authors should make it clearer that the E,E-8-mer.C60 is stable/favourable in solution and this has been established in the previous paper (reference 30). From the current study it isn't clear how stable this complex is and this would help the reader's understanding of the deposition process and whether it is surprising that the complex is deposited in the form described. I don't think it is surprising.

Indeed the results are not particularly significant in moving beyond what has been previously described by the same team. I don't find the paper to be a significant step beyond what they have already published (e.g. reference 30) and therefore I cannot see the justification for publishing the article in a journal such as Nature Communications. If there is a case it certainly hasn't been made in the manuscript and therefore I cannot recommend publication of the article.

Reviewer #3 (Remarks to the Author):

This contribution contains some interesting results related to the self-assembly of π - "expanded" macrocyclic oligothiophenes hosting fullerenes at the liquid-solid interface of a TMA-modified HOPG surface. Scanning tunneling microscopy and spectroscopy have been employed to analyze the characteristics of the ordered monolayers formed by the different components and molecular dynamic simulations have been as well performed. The manuscript lacks however of the degree of originality, importance, and interest, which is necessary for publication in Nature Communication. Concerning the validity of the conclusions extracted from the experiments, major and deep relevant doubts emerge and a comment follows below. I cannot recommend the manuscript for publication in Nature Communication.

The paper based on the adsorption of complex 3 onto the TMA-modified HOPG surface but no evidence of adsorption of the pristine complex can be obtain neither from the STS results nor from the STM images. Those show namely the adsorption of the macrocyclic oligothiophenes without fullerenes. The crystal structure of 3 bases on weak Van der Waals interactions and packing effects but is absolutely no proof for a stable complex formation at room temperature in heptanoic acid solution. The authors have to revise deeply the assumptions based on this preliminary key point over the entire manuscript. There is too much speculation about the results and data interpretation, which is not the best scientific approach that can be done with the experimental results presented in this communication. On another level, the authors claim several times about the need of "control" of the self-assembling and about supramolecular "engineering", but fairly speaking the paper show no more, no less, than basic observation of a process

We appreciate that the reviews expressed constructive and positive comments. In order to improve this work, corrections based on comments from the reviewers are included in the revised version of the manuscript and supplemental information. Please find below our detailed responses to all concerns raised by the reviewers.

Reviewer #1

This study is one of the very few reports on the supramolecular organization of multicomponent functional molecules. A self-assembled monolayer of trimesic acid at the liquid (1-heptanoic acid) – solid (graphite) interface acts as a template to host a complex of an oligothiophene macrocycle and C60. The TMA layer is necessary to direct the self-assembly of the oligothiophene macrocycle – C60 complex. The subsequent organization of this complex is ruled by the crystallinity of the TMA template layer. Molecular dynamics simulations confirm the tentative models based on the STM imaging. Furthermore, evidence for multilayer formation is provided. Interestingly, using scanning tunneling spectroscopy, space-resolve info is provided on the local electronic characteristics of this multicomponent system, showing rectifying behavior. A second layer of the macrocycle – C60 complex improves the rectifying behaviour.

While several aspects of this study have in one way or the other been reported before (not necessarily for the same system), e.g.

- Complexation of C60 and an oligothiophene macrocycle (Bauerle et al.)*
- STS on such complex of C60 and an oligothiophene macrocycle (Bauerle et al.)*
- STS on multicomponent systems (by Rabe et al.)*
- Multicomponent self-assembly involving trimesic acid (Lackinger et al., De Feyter et al).*
- Templating effect of trimesic acid*
- Multilayer formation involving C60 (Beton et al.)*

the combination provided in this study is of interest, of high quality, and complex. The data (STS) and data interpretation are supported by semi-quantitative modelling, based on concepts demonstrated by the Rabe group before.

While I favour publication, a number of aspects should be addressed in detail. In particular, more details should be given (in the supporting information).

Question/comment 1. *At this stage, not enough information is given to evaluate the statistical relevance of the STS data. In addition, certain statements seem not very well supported by the experimental evidence (e.g. difference in rectification ratio between 1 ML and 2 ML)*

Answer/correction 1. Regarding the statistical relevance of the STS data, we re-evaluated the data and calculated the average (arithmetic mean) of the rectification ratios, $R(V_0)$, at the maximum bias value V_0 for all the curves, instead of the previously formulated average rectification ratio, R_{av} . Changes have been made in the section “Results and discussion” of the manuscript and in the supplemental information.

Question/comment 2. *Concentration / composition of the solutions (for STS and STM topographs).*

Answer/correction 2. As mentioned in the method section, all solutions used heptanoic acid as solvent. Concentration has been added to the method section.

- Details on the STS

- **Question/comment 3.** *Fig. 3: what are the error bars for positive biases?*

Answer/correction 3. Error bars for all bias values are now displayed.

- **Question/comment 4.** *In general: What is the initial setpoint (current, voltage) to record the STS curves? (provide in Caption Figure 4, not in Figure 3).*

Answer/correction 4. Initial setpoint values for the STS curves are now in the revised version.

- **Question/comment 5.** *Fig. 3: Why a “saturation” at -1,2 V? What is meant by “saturation”? Do you mean “levels off”?*

Answer/correction 5. For bias values below -1.2 V the current levels off in the case of one templated monolayer of complexes. We called this saturation.

- **Question/comment 6.** *Fig. S7: are these averaged curves?*

Answer/correction 6. These curves are averaged as mentioned in the supplemental information. We added error bars for Fig. S7 in the revised version of supplemental information

- **Question/comment 7.** *Page 6, supporting info: what do you mean by “adequate average”?*

Answer/correction 7. The STS curves are averaged using a lognormal function to discard inappropriate data which could arise by fluctuations of the tip structure (Severin, N. *et al.* Data scattering in scanning tunneling spectroscopy. *Ultramicroscopy* **109**, 85 (2008)). The text is changed accordingly.

- **Question/comment 8.** *What about lateral drift effects during STS recording? In other words, how reliable are the claims concerning the exact position of the STS recording (e.g. center of the macrocycles, etc.)*

Answer/correction 8. In order to decrease the effect of lateral drift while recording STS, the tip was positioned over the region of interest as close as possible to the most recently scanned line. Typically, it takes about 25 ms to record a single STS curve and the lateral drift velocity is less than 0.1 nm/s. This explanation has been added to the method section.

- **Question/comment 9.** *In general, STS data presented: are these recorded in the same session? What about differences between sessions?*

Answer/correction 9. The STS data presented in each figure are from a single session. The tip to sample distance is unknown, it varies between sessions and it varies the current range, but not the characteristics of the I-V curves. For example: Fig. S8a and Fig. S9a show the STS data for the same system in different sessions, but the saturation (level-off) bias remains around -1.2 V and also the rectification ratio at the maximum bias.

- **Question/comment 10.** *Fig S9b: rectification ratio: based upon these data, it is dangerous to claim there is a significant difference in the rectification ratio between 1 ML and 2 ML, and the data are very scattered.*

Answer/correction 10. Fig S9 has been updated and error bars added. The procedure to calculate the average value of the rectification ratio has been changed as described in Answer/correction 1.

In addition, there are several minor typos that should be corrected.

Reviewer #2

The article by Rabe et al. builds upon a previous study (reference 30) which describes the synthesis and characterisation of the ‘Saturn-like’ complexes formed between oligothiophenes and C₆₀. In the current study a series of additional surface based characterisation is reported including STS etc.. The work is well carried out and it is difficult to criticise the quality of the science.

Question/comment 1. *The authors should make it clearer that the E,E-8-mer.C₆₀ is stable/favourable in solution and this has been established in the previous paper (reference 30). From the current study it isn't clear how stable this complex is and this would help the reader's understanding of the deposition process and whether it is surprising that the complex is deposited in the form described. I don't think it is surprising.*

Answer/correction 1. In the previous publication, we investigated the stability of the complex in toluene solution as part of its crystal characterization (Shimizu, H. *et al. J. Am. Chem. Soc.* **137**, 3877 (2015)). In the present study we use a different solvent, heptanoic acid, which might cause a different in-solution stability of the complex. However, we believe this is not a key issue for the final outcome of the adsorption process.

In fact, the formation of the self-assembled monolayer may follow two pathways:

- (1) Two-step adsorption: The E,E-8mer·C₆₀ complex is first dissociated in solution, and C₆₀ is adsorbed on TMA sites to afford **A**. Then, E,E-8-mer forms E,E-8mer·C₆₀ complex on TMA sites, and the complex forms a 2D network **B**.
- (2) One-step adsorption: The E,E-8mer·C₆₀ complex is adsorbed on TMA sites to form the bilayer self-assembly **B**.

These two pathways afford the templated bilayer self-assembly of the *E,E*-8mer·C₆₀ complex, although it is difficult to determine which process is favored. In the case of the first process, the stability of the *E,E*-8mer·C₆₀ complex in heptanoic acid is not a serious problem. In the Griessl *et al.* study (Griessl, S. J. H. *et al. J. Phys. Chem. B* **108**, 11556–11560 (2004)), C₆₀ is adsorbed on TMA sites to afford **A**, although with low density of adsorbed C₆₀ due to the low solubility of C₆₀ in heptanoic acid. Therefore, the low solubility of C₆₀ in heptanoic acid displaces the equilibrium towards the second process.

We have added a short discussion about the dynamics of the adsorption process in the section “Results and discussion” of the manuscript.

Question/comment 2. *Indeed the results are not particularly significant in moving beyond what has been previously described by the same team. I don't find the paper to be a significant step beyond what they have already published (e.g. reference 30)...*

Answer/correction 2. In the present manuscript we report on the formation of a highly complex target structure, namely a templated bilayer of the *E,E*-8mer·C₆₀ complex. This goes significantly beyond the self-assembly of the sole *E,E*-8mer macrocycle. In fact the complex does not assemble by itself on graphite, it requires a specific molecular template (TMA). In order to achieve this goal we had to combine:

1. Modification of the HOPG surface using a TMA network
2. Effective (high density) host-guest complexation of C₆₀ into TMA adsorption sites.
3. Epitaxial self-assembly of the *E,E*-8mer·C₆₀ complex.
4. Further self-assembly of a second monolayer of complexes on top of the first one with a similar arrangement to its crystal phase.
5. Verification of the templated mono and bilayer by bias dependent imaging, STS and MD simulations.

... and therefore I cannot see the justification for publishing the article in a journal such as *Nature Communications*. If there is a case it certainly hasn't been made in the manuscript and therefore I cannot recommend publication of the article.

Reviewer #3

*This contribution contains some interesting results related to the self-assembly of π -“expanded” macrocyclic oligothiophenes hosting fullerenes at the liquid-solid interface of a TMA-modified HOPG surface. Scanning tunneling microscopy and spectroscopy have been employed to analyze the characteristics of the ordered monolayers formed by the different components and molecular dynamic simulations have been as well performed. The manuscript lacks however of the degree of originality, importance, and interest, which is necessary for publication in *Nature Communication*. Concerning the validity of the conclusions extracted from the experiments, major and deep relevant doubts emerge and a comment follows below. I cannot recommend the manuscript for publication in *Nature Communication*.*

Question/comment 1. *The paper based on the adsorption of complex 3 onto the TMA-modified HOPG surface but no evidence of adsorption of the pristine complex can be obtain neither from the STS results nor from the STM images. Those show namely the adsorption of the macrocyclic oligothiophenes without fullerenes.*

Answer/correction 1. Our experiments using bias dependent STM imaging showed two *kinds* of patterns for different bias values: in the experiments with one monolayer of complexes on TMA, we resolve the TMA honeycomb structure for negative sample bias, and macrocycles for the opposite bias.

Macrocycle and C₆₀ form a donor-acceptor complex where the HOMO (LUMO) is located in the macrocycle (C₆₀). Using a model that has been previously described (Seifert, C. *et al. Phys. Rev. B* **80**, 245429 (2009)), we interpret the invisibility of fullerene C₆₀ to the fact that the LUMO energy cannot be reached within our accessible bias range.

Furthermore, we performed experiments using TMA and macrocycle without C₆₀ and never got the double layer structure (detailed in the Supplemental Information section); therefore we conclude fullerene C₆₀ plays a decisive role in the formation of the double layer.

Finally, the STS measurements of *E,E*-8mer·C₆₀ complex on TMA-modified HOPG have features only reported before on fullerenes C₆₀ adsorbed on thiophene macrocycles (Mena-

Osteritz, E. & Bäuerle, P. *Adv. Mater.* **18**, 447 (2006)), supporting further that the C₆₀ is present and plays a role in the adsorption of the pristine complex on the template.

Question/comment 2. *The crystal structure of 3 bases on weak Van der Waals interactions and packing effects but is absolutely no proof for a stable complex formation at room temperature in heptanoic acid solution. The authors have to revise deeply the assumptions based on this preliminary key point over the entire manuscript.*

Answer/correction 2. We already addressed this issue above with the **Answer/correction 1** to the reviewer 2.

Question/comment 3. *There is too much speculation about the results and data interpretation, which is not the best scientific approach that can be done with the experimental results presented in this communication.*

Answer/correction 3. As pointed out in **Answer/correction 1**, we have provided multiple arguments to support one of our conclusions.

Question/comment 4. *On another level, the authors claim several times about the need of “control” of the self-assembling and about supramolecular “engineering”, but fairly speaking the paper show no more, no less, than basic observation of a process.*

Answer/correction 4. Our results are indeed about a supramolecular engineering: we observed that the *E,E*-8mer·C₆₀ complex, contrary to the pure *E,E*-8mer macrocycle, did not self-assemble on bare graphite. We, therefore, design a suitable molecular template (TMA), which allows to control the self-assembly of the complex on the surface.

With kind regards

Jürgen Rabe (for all authors)

REVIEWERS' COMMENTS:

Reviewer #1 (Remarks to the Author):

The authors have addressed my comments in the way I expected they would do. So, I'm pleased with the changes made, and I don't have arguments to change my initial positive evaluation of this manuscript.

Reviewer #2 (Remarks to the Author):

I find the paper to be improved following the revision process. I am still slightly unconvinced by the level of originality in the paper but the authors have made attempts to address the issues raised by all three reviewers.

I therefore am happy to recommend acceptance of the article.